# Chromosome Comparisons of Australian *Scaptodrosophila* Species

**DOI:** 10.3390/insects13040364

**Published:** 2022-04-07

**Authors:** Ann Jacob Stocker, Michele Schiffer, Eduardo Gorab, Ary Hoffmann

**Affiliations:** 1Pest and Environmental Adaptation Research Group, School of Biosciences, Bio21 Institute, University of Melbourne, Parkville, VIC 3010, Australia; ary@unimelb.edu.au; 2Daintree Rainforest Observatory, James Cook University, Cape Tribulation, QLD 4873, Australia; michele.schiffer@jcu.edu.au; 3Departamento de Genética e Biologia Evolutiva, Instituto de Biociências, Universidade de São Paulo, Rua do Matão 277, Cidade Universitária, São Paulo 05508-090, SP, Brazil; gorabcrow@gmail.com

**Keywords:** *Scaptodrosophila*, karyotypes, heterochromatin, NORs, comparison, evolution

## Abstract

**Simple Summary:**

*Scaptodrosophila* are a diverse group of flies that are widespread in Australia but have received limited research attention. In this study, we characterized the chromosomes of 12 *Scaptodrosophila* species. We found that the structural changes in their chromosomes are very similar to those seen in *Drosophila*, a related genus of flies. This includes the amplification of repetitive elements and changes in chromosome length in small (dot) chromosomes and the sex chromosomes. We also found numerous weak points along the arms of polytene chromosomes, which suggest the presence of internal repetitive sequences. Regions producing the nucleolus are at the same chromosome positions in *Scaptodrosophila* and *Drosophila*. These chromosomal studies provide a foundation for future genetic studies in *Scaptodrosophila* flies.

**Abstract:**

The *Scaptodrosophila* represent a diverse group of *Diptera* closely related to *Drosophila.* Although they have radiated extensively in Australia, they have been the focus of few studies. Here, we characterized the karyotypes of 12 *Scaptodrosophila* species from several species groups and showed that they have undergone similar types of karyotypic change to those seen in *Drosophila*. This includes heterochromatin amplification involved in length changes of the sex and ‘dot’ chromosomes as well as the autosomes, particularly in the *coracina* group of species. Numerous weak points along the arms of the polytene chromosomes suggest the presence of internal repetitive sequence DNA, but these regions did not C-band in mitotic chromosomes, and their analysis will depend on DNA sequencing. The nucleolar organizing regions (NORs) are at the same chromosome positions in *Scaptodrosophila* as in *Drosophila,* and the various mechanisms responsible for changing arm configurations also appear to be the same. These chromosomal studies provide a complementary resource to other investigations of this group, with several species currently being sequenced.

## 1. Introduction

The genus *Scaptodrosophila* is estimated to have diverged within the drosophilid lineage during the Cretaceous period about 70 million years ago. Further diversification occurred from the Eocene through the Holocene, about 50–0.7 million years ago [1]. The *Scaptodrosophila* were originally considered a subgenus of *Drosophila* and called *Pholadoris* in earlier studies [2,3], but in a taxonomic revision of the family, Grimaldi [4] elevated the group to genus rank, which was supported by molecular data from DeSalle [5].

Of the 36 drosophilid genera represented in Australia, approximately a third of the described species belong to the genus *Scaptodrosophila*. This prevalence of *Scaptodrosophila* in the Australasian drosophilid radiation is unique when compared to diversification elsewhere [6,7]. However, the genus has been the subject of relatively few scientific studies [8,9]. The developmental requirements of most *Scaptodrosophila* species are unknown and appear to be restrictive since few species are widespread in urban environments [6]. *Scaptodrosophila* species have been reported as feeding or breeding on tree sap, fungi, fruit and flowers [8,10]. Unlike *Drosophila* species, most cannot be collected on fruit baits, except for members of the *coracina* species group [11]. *Scaptodrosophila hibisci* and *S. aclinata* breed in the flowers of native *Hibiscus* species [7,12]. There are also species that feed on and form galls in the stems of a bracken fern [13], while others feed on eucalyptus leaf litter in the crowns of tree ferns [8]. Species from the genus may have important ecological functions such as in pollination and nutrient cycling, as well as in acting as a food source for other organisms.

Some *coracina* group species can be reared on conventional *Drosophila* laboratory medium [9]. However, other species must be supplied with fruit or flowers to stimulate egg laying. In the laboratory, mature larvae usually complete development by leaving their food vial and burrowing into surrounding sand or vermiculite to pupate [8]. Their ‘skipping’ or springing behavior facilitates this and likely aids their dispersal in nature. The practice of burrowing into ground material to pupate may have evolved as a mechanism to protect the quiescent stage from environmental extremes and fires as well as predation and parasitism. It is not surprising, therefore, that most species have been difficult to maintain in the laboratory, particularly in long-term culture [8].

Given the difficulties in capturing and maintaining *Scaptodrosophila* species, few chromosomal studies have focused on *Scaptodrosophila,* with the most complete study carried out by Bock [11] on six members of the *lativittata* species complex of the *coracina* group. An early study by Mather [3] presented a scheme of chromosome evolution. However, he considered them a subgenus of *Drosophila*, and his diagram of chromosome evolution mixed *Drosophila* and *Scaptodrosophila* (*Pholadoris*) species. A more recent study by Wilson et al. [12] on *S. hibisci* indicated the presence of male recombination and neo-sex chromosome formation in this species.

Here, we established conditions for the short-term culture of several *Scaptodrosophila* and used these lines to karyotype some species. Since *Scaptodrosophila* and *Drosophila* species have been separated for about 70 myr and adult morphology is still quite similar, we tested if the karyotypic changes that had occurred in species from these groups were also similar. The cerebral ganglion chromosomes of the species that we managed to culture were examined morphologically and also subjected to C-banding for constitutive heterochromatin localization and fluorescence in situ hybridization (FISH), to locate ribosomal RNA genes. We attempted silver staining to examine the location of active nucleolar organizing regions (NORs). We examined the polytene chromosomes but did not attempt a detailed banding comparison.

## 2. Materials and Methods

### 2.1. Species Examined

Species were mainly collected in northeastern Australia and identified by one of us (M.S.) (Table 1). Several species are either new records for Australia or undescribed, and their classification is currently under investigation.

Isofemale lines were established in the laboratory for the collected species. Although most of the *Scaptodrosophila* studied here have white larvae, larvae of *S. novoguineensis* and *S.* sp. *aff. novoguineensis* become purple as they age. This purple pigment occurs in both the fat body and hemolymph but appears to originate from the fat body. All feeding larvae had long posterior spiracles and showed the characteristic skipping behavior. The species that we studied preferred pupating in vermiculite that surrounded the culture vials. However, we cultured two species, *S.* sp. *aff. concolor* strain CBN17 and *S. lativittata,* for a period solely in food vials, and *S. xanthorrhoeae* can also be maintained indefinitely without access to a vermiculite substrate.

Polytene chromosomes of *S. nitidithorax* were examined for inversion polymorphism. The mitotic karyotype of this species comes from Bock [11]. Mitotic and polytene chromosomes of a small number of *S. specensis* had also previously been examined by Bock [11]. We C-banded several ganglion chromosome preparations of *S. specensis.* Polytene and mitotic chromosomes were examined for *S. bryani* and *S.* sp. *aff. concolor* strain CBR1. These techniques along with C-banding were completed for *S. novoguineensis*, *S. lativittata* and *S. evanescens*. All methodologies plus NOR localization were completed for *S. xanthorrhoeae, S. claytoni, S. cancellata, Scaptodrosophila* sp. *aff. cancellata, S.* sp. *aff. concolor* strain CBN17 and *Scaptodrosophila* sp. *aff. novoguineensis*.

The chromosome techniques used were the following: squash preparations of salivary gland polytene chromosomes in lacto-aceto orcein, squashes or drop spreads of brain mitoses stained with lacto-aceto orcein or Giemsa (Gibco, Adelaide, Australia), C-banded spreads of brain mitoses stained with Giemsa, NOR silver staining of mitotic spreads and fluorescence in situ hybridization (FISH) to identify the NORs (Table 2).

### 2.2. Polytene Chromosome Squashes

Glands were removed in *Drosophila* Ringer’s solution and fixed in 50% acetic acid for 7–10 s and 1N HCl for about 30 s. They were then moved to a small drop of mounting medium (60% acetic acid: lactic acid, 1:1) on a siliconized coverslip and allowed to clear before a very small amount of lacto-acetic orcein stain was added and the coverslip picked up with a cleaned slide and tapped/squashed to spread the chromosomes. To visualize the nucleolus, some preparations were fixed in 3:1 ethanol:acetic acid or methanol:acetic acid.

### 2.3. Cerebral Ganglion Chromosome Spreads

The cerebral ganglions were removed in *Drosophila* Ringer’s solution, placed in 0.5% Na citrate for 10–15 min to spread the chromosomes and then fixed in several changes of methanol:acetic acid:water (11:11:1) for 1–2 min or longer. Tissue was then stained in lacto-acetic orcein on a siliconized coverslip for 10 min, a drop of 60% acetic acid was added and the preparation was picked up with a cleaned slide and squashed. For slides to be used in C-banding, NOR silver staining or FISH, tissue was either squashed in 60% acetic acid and the coverslip removed after freezing in liquid nitrogen, or the cells were allowed to disassociate in 60% acetic acid and small volumes picked up with a micro-pipette, dropped on a clean, warmed slide and air dried.

### 2.4. C-Banding

C-banding generally followed the technique used by Bedo [14] with a few variations. We used a 2–3 min incubation in 5% Ba(OH)_2_ at 50 °C followed by extensive washing in running tap water and deionized water. The Ba(OH)_2_ did not go completely into the solution, and extensive washing in warm tap water was important for getting rid of some of the precipitate. This was followed by air drying, incubation for 1 h in 2X SSC (saline sodium citrate) buffer at 60 °C and a rinse in deionized water. The slide was then either stained immediately in 4% Giemsa (Gibco) for 15–30 min, or dried and stained the following day. This was followed by a quick rinse in deionized water. After drying well, the preparation was mounted in Gurr’s Neutral Mounting Media and examined.

### 2.5. NOR Silver Staining

NOR silver staining was based on the technique of Howell and Black [15]. Since staining on mitotic chromosome figures was not consistent, various modifications of temperature and time [16,17,18], pH [19], pre- and post-treatment of slides [20,21,22,23,24] and age of the slide were tested. None were satisfactory in terms of giving consistent results. We tried heating in a microwave oven [25], but our preparations tended to be unevenly developed with no consistent deposition of silver grains over a particular chromosome.

### 2.6. Fluorescence In Situ Hybridization (FISH)

Fluorescence in situ hybridization was carried out using two methods. The same result was achieved by both. 1. A *Drosophila melanogaster* DNA clone, pDm238 in pBR322 [26], was labeled by nick translation and used as a probe as described previously [27]. This probe was an 11.5 kb fragment that contained the 18S, 5.8S and 28S genes of *D. melanogaster* plus an intergenic spacer. 2. Total *Drosophila* RNA was used as a probe following the technique of Madalena et al. [28] or a slightly modified version of this technique. Approximately 40 *Drosophila melanogaster* flies were etherized and immediately frozen in liquid nitrogen. They were homogenized in 0.8 mL Trizon (Invitrogen, Adelaide, Australia), and chloroform was added to a total volume of 1 mL with mixing by Eppendorf inversion. After 15 min at 4 °C, the preparation was centrifuged at 12,000 rpm at 10 °C for 10 min. One volume of frozen isopropanol was added to the extracted liquid phase with mixing, and the Eppendorf was kept in the freezer for 5 h. To recover the RNA, centrifugation was repeated as above and the pellet resuspended in formamide containing poly-(r)U (Sigma, Adelaide, Australia) (1 µg/mL).

Chromosome spreads were treated with RNase A diluted in 2X SSC (0.2 mg/mL) for 3 h at room temperature. The slides were washed in 0.5X SSC, and chromosome DNA denaturation was performed by one of two techniques. In the first, slides were treated as in Madalena et al. [28]. A probe mixture (6 µL) of 50% formamide, 2X SSPE (saline sodium phosphate EDTA buffer), 0.1% SDS and 100 ng of insect RNA mixed with poly (r)-U was applied to each air-dried slide and covered with a plastic coverslip. The slides were steam heated at 75 °C for denaturation and then kept in a closed plastic box at 37 °C overnight or longer for hybridization. In the second technique, slides were incubated in 0.08 N NaOH for 1 min at room temperature and then washed in 0.5X SSC and kept in ice-cold ethanol until the hybridization procedure was carried out. The slides to be hybridized were dried, and the hybridization mixture, consisting of 50% formamide containing RNA and 3X SSC, was applied under a plastic coverslip. The slides were incubated overnight at 37 °C.

Detection of the DNA-labeled probe was described in Stocker et al. [27]. The RNA probe was detected by incubation with goat IgG anti-RNA.DNA hybrid [29] and detected with rabbit IgG anti-goat labeled with TRITC (Sigma), as described in Madalena et al. [28] and Gorab et al. [30]. Slides were counterstained with 4′, 6-diamino-2-phenylindole (DAPI). They were mounted in Vectashield anti-fading solution (Vector Labs, Adelaide, Australia) and inspected with epifluorescence optics using Zeiss and Olympus microscopes equipped with CCD cameras and image analysis software.

## 3. Results

### 3.1. Lacto-Acetic Orcein Staining

The mitotic karyotypes of the 12 *Scaptodrosophila* species are shown in Table 3. We have also included karyotypes of the other *coracina* species whose chromosomes were examined by Bock [11] as well as those of *S. hibisci* studied by Wilson et al. [12]. Most of the species examined are members of the *coracina* group. Two have tentatively been placed in the *barkeri* group, one is in the *bryani* group and the rest are not currently grouped with other species. *Scaptodrosophila novoguineensis* and *S.* sp. *aff. novoguineensis* are different species but very similar morphologically.

The karyotypes of nine species in the *coracina* group are very similar, particularly with respect to their autosomes. The tenth species, *S.* sp *aff. cancellata*, currently undescribed, lacks the metacentric autosome and instead has two acrocentrics (rods). The major difference among these karyotypes is in the shape of the ‘dot’ and sex chromosomes. ‘Dot’ chromosomes range from very small ‘dots’ (*claytoni*, *evanescens*, *specensis*) to large metacentrics (*enigma*). The X and Y chromosomes also have variable shapes from acrocentrics to submetacentrics.

*Scaptodrosophila* sp. *aff. concolor* strain CBN17 has a superficially similar karyotype to the species in the *coracina* group, with a submetacentric X chromosome and a somewhat elongated ‘dot’ chromosome (Table 3, Figure 1A insert). The Y mainly appears as an acrocentric. A ‘dot’ chromosome is clearly apparent in the polytene set and could often be distinguished by its proximity to the nucleolus (Figure 1A). Polytene chromosomes in the *S.* sp. *aff. concolor* strain CBN17 lines that were examined are also polymorphic for a large terminal inversion on one of the autosomes (Figure 1A,B). The other end of this autosome shows pairing between repetitive sequences which is common in *Scaptodrosophila* (Figure 1B). Two additional polymorphic inversions were observed in the original isofemale line of *S.* sp. *aff. concolor* strain CBN17.

The other species that we studied showed a variable number of arm fusions. *Scaptodrosophila* sp. *aff. concolor* strain CBR1 has two pairs of metacentric autosomes and one pair of acrocentric autosomes. The ‘dot’ and X chromosomes have short arms, and the Y often shows a central constriction (Figure 2A). *Scaptodrosophila* sp. *aff. concolor* strain CBN17 and *S.* sp. *aff. concolor* strain CBR1 have been tentatively placed in the *barkeri* group taxonomically, due to their superficial similarity to *S. concolor*, but this is likely to be revised upon examination of these species compared with type specimens of *S. concolor*, particularly in light of their chromosomal differences. The chromosomes of *S.* sp. *aff. concolor* strain CBR1 more closely resemble those of *S. xanthorrhoeae* (Figure 2B), which has not been assigned to a group. However, *S. xanthorrhoeae* has an acrocentric X and Y and small ‘dots’. *Scaptodrosophila novoguineensis* and *S.* sp. *aff. novoguineensis* show further reductions, having two metacentric autosome pairs, an acrocentric-shaped X and Y and small ‘dots’. *Scaptodrosophila bryani* has the most numerically reduced karyotype with two metacentric chromosome pairs, an acrocentric X and Y and no ‘dot’.

*Scaptodrosophila hibisci* [12] has a complex karyotype that appears very different from the *Scaptodrosophila* karyotypes that we have examined here. It has a large metacentric chromosome pair with variable Giemsa staining of homologues, three small metacentric pairs, a small acrocentric and a small ‘dot’ chromosome pair. Experiments by Wilson et al. suggested neo-sex chromosome formation in this species [12]. While sex chromosomes have not been definitively identified, the authors suggest that the large metacentrics could represent sex chromosomes, although additional translocations could have brought other chromosomes into the sex-determining system [12].

The quality of the polytene spreads in the species we examined varied, and good spreads proved difficult to obtain for most species. *Scaptodrosophila* chromosomes have numerous ‘weak points’ which are prone to breakage and the extrusion of small chromosome segments. This made it difficult to define individual chromosomes and distinguish the ‘dot’ chromosome from small segments of other chromosomes that had become detached. An extreme example is shown in the squash of an *S. bryani* nucleus (Figure 3A). Besides *S.* sp. *aff. concolor* strain CBN17, possible dots could be observed in polytene sets of *S. cancellata* (Figure 3B) and *S.* sp. *aff. cancellata* which both had metacentric dot chromosomes in mitotic spreads. ‘Dot’ chromosomes in polytene sets of other species were not repeatedly observed.

### 3.2. C-Banding

C-banding was carried out to examine the role of heterochromatin addition in karyotype changes among these *Scaptodrosophila* species. Some of the species examined by Bock [11] had large ‘dot’ chromosomes. Enlarged ‘dot’ chromosomes were also observed for several of the species in our investigation. We would expect these chromosomes to be C-band-positive. We were also interested in examining other chromosomes for the addition of C-banded heterochromatin regions. C-banding of ganglion cell chromosomes of the six *coracina* group species that were available gave the results shown in Figure 4. The three species with very small ‘dot’ chromosomes, *S. claytoni*, *S. specensis* and *S. evanescens*, have a similar C-banding pattern (Figure 4A). C-banded heterochromatin is consistently located at either side of the centromere of the metacentric autosomes and at the centromeric end of the acrocentric autosomes and the X chromosome. The Y chromosome is entirely C-band-positive. Occasionally, internal C-bands were observed, such as those on the X chromosome of *S. claytoni*, but these were not always present. Surprisingly, the ‘dot’ chromosomes of these three species are not always C-band-positive. Three species have larger, metacentric dot chromosomes. Except for its metacentric, C-band-positive ‘dot’ chromosome, *S. lativittata* has a C-banding pattern similar to that described for the species above. *Scaptodrosophila cancellata* has more C-banding heterochromatin at the centromeric ends of its acrocentrics than the other species. Both *S. cancellata* and *S.* sp. *aff. cancellata* have submetacentric X chromosomes with C-band-positive short arms and prominent C-band-positive metacentric ‘dot’ chromosomes (Figure 4B,C). *Scaptodrosophila* sp. *aff. concolor* strain CBN17 (Figure 4D) is not a member of the coracina group but, like them, has a pair of metacentric autosomes with C-banding near the centromere and C-bands near the centromeric end of the three acrocentric autosomes. However, the ‘dot’ chromosomes of this species have one part somewhat more darkly stained than the other, and the submetacentric Xs have only a fine C-band at the region of the presumed centromere (Figure 4D).

*Scaptodrosophila xanthorrhoeae* and *S*. sp. *aff. concolor* strain CBR1 have similar karyotypes (Table 3), but *S.* sp. *aff. concolor* strain CBR1 was lost before C-banding had been carried out. The C-banded karyotype of *S. xanthorrhoeae* is shown in Figure 4E. C-banded heterochromatin is observed at either side of the centromere of the metacentric chromosomes and at the centromeric ends of the acrocentrics. The ‘dot’ chromosome is C-band-positive in many, but not all, sets. *Scaptodrosophila* sp. *aff. concolor* strain CBR1 (Figure 3A) has a small, metacentric ‘dot’ chromosome which often stains darkly in orcein squashes, similar to the long, apparently metacentric Y chromosome. This likely indicates that they would be C-band-positive.

*Scaptodrosophila novoguineensis* has the same karyotype as *S.* sp. *aff. novoguineensis*. C-banding is shown for *S.* sp. *aff. novoguineensis*, a new species which is currently being described (Figure 4F). C-band-positive heterochromatin is located at either side of the centromere of the two metacentric chromosomes and at the centromeric end of the X chromosome. The Y chromosome is totally C-band-positive. The dot chromosome is positive in this preparation.

In these *Scaptodrosophila* species, no consistent internal C-bands in mitotic chromosomes were detected with the technique used, although the number of weak points and inverted repeats in polytene chromosomes suggested the presence of internal repetitive DNA.

C-banding was conducted by Wilson et al. [12] for cerebral ganglion chromosomes of *S. hibisci* and is shown for one of the species’ two chromosome 1 forms [12]. Considerable C-banded heterochromatin is located on all chromosomes, including entire arms of chromosomes 2 and 3 and almost an entire arm of chromosome 1. It is clear that heterochromatin additions and deletions have been important in shaping the karyotype of this species, perhaps changing acrocentrics to metacentrics, but further study is required to decipher the changes that have occurred.

### 3.3. NOR Location

The location of the NOR was examined in *S. cancellata*, *S.* sp. *aff. cancellata*, *S. claytoni*, *S.* sp. *aff. concolor* strain CBN17, *S. xanthorrhoeae* and *S. sp. aff. novoguineensis*. For all species except *S. sp. aff. concolor* strain CBN17, in situ hybridization located the ribosomal DNA genes on the sex chromosomes. In *S. xanthorrhoeae* (Figure 5A,B), *S. claytoni* and *S.* sp *aff. novoguineensis*, one NOR was at the very end of the short heterochromatic block on the acrocentric X, probably just above the heterochromatin. *Scaptodrosophila cancellata* and *S.* sp. *aff. cancellata* had a submetacentric X, with the short arm almost completely heterochromatic (Figure 4B,C). In both species, the NOR was at the end of this arm (Figure 5C–F). In all of these species except *S. claytoni*, we found another NOR at one end of the Y chromosome (Figure 5B,D,F). For *S. claytoni*, we could not ascertain if the Y chromosome contained an rDNA site because we never managed to find a male included among our in situ slides. For *S. lativittata*, which was not considered in the in situ hybridizations, the presence of distinctive satellites at the heterochromatic end of the X chromosomes indicated that this was one of the NOR sites (A. Stocker, personal observations). In several species, there appeared to be variation in the number of ribosomal cistrons between the two X chromosomes, but this needs further investigation.

*Scaptodrosophila* sp. *aff. concolor* strain CBN17 differed from the other species in having its ribosomal genes on the dot chromosome (Figure 6A–C). There was very little heterochromatin present on the X chromosome of this species (Figure 4D), and neither the X nor Y showed any rDNA.

Attempts to identify the site of the active nucleolus in mitotic spreads using variations of the silver staining technique of Howell and Black [15] were mostly unsuccessful. The only consistent silver staining was at the presumed nucleoli of interphase cells (Figure 7A1). Occasionally, a silver-stained nucleolus or fragment of a nucleolus could be seen attached to a prophase chromosome, but it was usually unclear which chromosome was involved (Figure 7A1). When staining occurred on metaphase chromosomes, it was usually at centromeres. Rarely, staining occurred at a chromosome region that we knew from FISH results contained rDNA (Figure 7B). Most mitoses did not show silver staining at all. Such variable results have also been observed in other species [22,31,32].

## 4. Discussion

### 4.1. C-Banding

C-banding identifies constitutive heterochromatin blocks which are more resistant than euchromatin to DNA removal [33]. The C-banding pattern of the *Scaptodrosophila* chromosomes was generally consistent (Figure 8), with the exception of the smaller ‘dot’ chromosomes which sometimes did not C-band. The heterochromatic nature of the ‘dot’ chromosomes is a conserved feature throughout the drosophilids. However, their repeat level is considerably lower than pericentric heterochromatin since other gene sequences are dispersed within them [34]. Since the genes in ‘dot’ chromosomes of different species are apparently the same, smaller ‘dot’ chromosomes should contain fewer repeats than larger ‘dots’ and therefore be less heterochromatic. This could at least partially explain their lack, or variability, of C-banding in this study. In all the species we studied, the Y chromosome C-banded very strongly. It was usually impossible to discern clear structural variation along this chromosome, although sometimes there were constrictions or a region that looked darker than the rest. If the strength of C-banding is related to a high repeat level, the Y chromosome DNA must be very highly repetitious. There were no consistent C-bands within the arms of the mitotic chromosomes despite the presence of numerous weak points, indicating internal heterochromatin along the polytene arms.

The extensive C-banded regions in *S. hibisci* [12] suggest that the distinctive karyotype of this species may have been obtained by heterochromatin addition to acrocentrics, changing them to metacentrics or submetacentrics. The authors suggest that the rearranged banding between the chromosome 1 homologues is probably caused by a pericentric inversion. Chromosome 1 is seen in four forms and two combinations. The forms with the end satellite have characteristics of the X chromosome, with the satellite indicating the position of the NOR, which is commonly found on the X in drosophilids. The thin, intermediately banded chromosome that it is paired with in Type 1 cells is suggested by the authors to be a partially heterochromatized neo-Y chromosome. C-banding is not shown for this chromosome 1 homologue, but if the pale Giemsa staining corresponds to the absence of C-bands, the weak staining of this homologue may mean that other factors in addition to type and number of repeats are responsible for the presence of C-banding. Translocations between crucial Y regions and other chromosomes may also have played a role in the staining differences.

### 4.2. NOR Location

Most species in the genus *Drosophila* that have been examined thus far have NORs on their sex chromosomes, and this is believed to be the ancestral condition [35]. NOR-associated sequences probably assist in pairing between the sex chromosomes during meiosis [36,37]. On the *Drosophila* X chromosomes, NORs were usually located near the centromere at heterochromatic regions [35]. This was also the case in three of the *Scaptodrosophila* species for which the NOR position was examined (Figure 8). In *Drosophila hydei*, however, the X is metacentric and the NOR is located near the distal end of its heterochromatic arm [38]. This position is analogous with the results we obtained in *S. cancellata* and *S.* sp. *aff. cancellata* (Figure 8) and suggests that heterochromatin amplification between the centromere and the NOR moved this region toward the end of the arm during species evolution. The Y chromosome is entirely heterochromatic in *Drosophila* species, and NORs on this chromosome are more widely distributed [35]. In the group of *Scaptodrosophila* species examined, the NOR was always observed at one end of the Y (Figure 8). Some species in the genus *Drosophila* have NORs on chromosomes other than the sex chromosomes. Most of the members of the *D. ananassae* group have a NOR site on the ‘dot’ chromosome [35]. One of the *Scaptodrosophila* species that we examined, *S.* sp. *aff. concolor* strain CBN17, also has its NOR on the dot chromosome (Figure 8). Two Hawaiian *Drosophila* species have a NOR on a different, non-heterochromatic autosome (the B element according to Mueller’s [39] nomenclature), but the band at which it is located is heterochromatic [40]. Despite the changes in the NOR position, it is consistently associated with heterochromatin regions.

### 4.3. Karyotype Evolution

In *Drosophila*, karyotype examination [39,41] and more recent sequencing [42] have given convincing evidence that the primitive chromosome configuration is five pairs of acrocentrics (rods), including sex chromosomes, and one pair of ‘dots’, which are usually very small and heterochromatic. When the chromosome number differed from this, gene comparisons usually could identify six representative ‘elements’ called A–F, with A being the X chromosome and F the ‘dot’ chromosome. The *Scaptodrosophila* species in the current study have chromosome numbers ranging from five pairs of acrocentrics, a submetacentric X chromosome pair and a pair of ‘dots’, to two metacentric pairs, acrocentric sex chromosomes and no ‘dots’. Using arm changes and heterochromatin variation, we can suggest how some of these karyotypes originated from a hypothetical primitive one.

The most common karyotype among the species that we have examined is one pair of metacentrics and three pairs of acrocentric autosomes, acrocentric or submetacentric sex chromosomes and ‘dot’ chromosomes of varying sizes. This karyotype is found in all except one member of the *coracina* group species and also in *S.* sp. *aff. concolor* strain CBN17. It can be derived from the primitive karyotype by postulating a pericentric inversion in an acrocentric autosome that moved the centromere to a central position, pericentric inversions and heterochromatin amplification in some of the sex chromosomes and heterochromatin amplification in ‘dot’ chromosomes (Figure 9). The exceptional karyotype among the *coracina* species is *S.* sp. *aff. cancellata*, which has five pairs of acrocentric autosomes, submetacentric sex chromosomes and large metacentric ‘dot’ chromosomes. This karyotype could have been obtained by dissociation of the metacentric autosome into two acrocentrics. Heterochromatin changes are very similar to those in *S. cancellata.* Since other members of the *coracina* group have an acrocentric X [11], and in *S. claytoni*, the NOR is associated with centromeric heterochromatin, it is most probable that the heterochromatic arm of the X developed through extension of heterochromatin between the centromere and the NOR, leaving the NOR positioned near its end. Changes in the length of ‘dot’ chromosomes were common in the *coracina* species examined by Bock [11] (Figure 1, Figure 2, Figure 3, Figure 4, Figure 5 and Figure 6).

*Scaptodrosophila* sp. *aff. concolor* strain CBN17 was tentatively placed in the *barkeri* species group because of its superficial similarity to *S. concolor*, but its karyotype looked similar to that of the *coracina* species until the NOR sites were located. These were found to be on the ‘dot’ chromosomes rather than on the X and Y (Figure 10). The yellow/brown bodies and reddish/orange eyes of the adult flies were also quite different from the darker color of the *coracina* species that we examined. The changes that formed the *S.* sp. *aff. concolor* strain CBN17 karyotype are hypothesized to have been the same pericentric inversion described above, and transposition of the ribosomal DNA cistrons from the X to the dot chromosome as a pericentric inversion was formed in the X (Figure 10). In this process, the X appears to have lost most of its heterochromatin. It is possible that the rDNA on the dot chromosome came from the Y, since Y rDNA cistrons are also absent. To resolve this question, we would need an intermediate situation with a NOR on the dot chromosome and another on one of the sex chromosomes. With respect to its karyotype, this species appears to have diverged directly from the ancestor of the *coracina* group (Figure 10).

The other species show a reduction in chromosome number and changes in chromosome shape, with elongation of chromosome short arms in *S.* sp. *aff. concolor* strain CBR1, also tentatively placed in the *barkeri* species group (Figure 11). However, *S.* sp. *aff. concolor* strain CBR1 looks chromosomally similar to *S. xanthorrhoeae* (Figure 11), with a reduction in acrocentric autosomes to one and the gain of a smaller metacentric. A pericentric inversion could have formed the smaller metacentric, and a centromeric fusion between the two acrocentrics would form the larger acrocentric. Although we do not have C-banding for *S.* sp. *aff. concolor* strain CBR1, its short arms, not present in *S. xanthorrhoeae,* and larger metacentric ‘dot’ chromosomes are probably heterochromatic.

Further arm reduction is observed in *S. novoguineensis* and *S.* sp. *aff. novoguineensis,* both having karyotypes composed of two pairs of metacentrics, a pair of dots and an acrocentric X chromosome. The most straightforward manner that these changes could have evolved from a 5R, 1D karyotype is for two centric fusions to have occurred between four pairs of autosomes (Figure 11). Finally, *S. bryani* has lost the ‘dot’ chromosome, most likely by fusion with one of the longer chromosome arms. The absence of a ‘dot’ chromosome has been observed in several *Drosophila* species groups. In *D. busckii* [43], a basal *Drosophila* species, ‘dot’ chromosome genes are located near the heterochromatin of the X chromosome and also at a homologous region of the Y chromosome [44]. In situ hybridization also placed ‘dot’ chromosome genes of *D. lebanonensis* near the centromere of the acrocentric X chromosome (A element) [45]. Since X and ‘dot’ chromosomes have some similar characteristics, it has been suggested that, in the ancient history of drosophilids, the ‘dot’ chromosome (F element) was part of the X chromosome [34]. In fact, sequencing and mapping of non-*Drosophila* Diptera have shown that the Mueller A element is an autosome and the ‘dot’ chromosome is the female sex chromosome in these outgroup species [46]. Based on these results, we might hypothesize that ‘dot’ genes would be translocated near the centromere of the X chromosome in *S. bryani*. However, in six species of the willistoni species group, ‘dot’ chromosome genes were located near the centromere of the E element, also acrocentric [47]. Since the X and Y are the only acrocentrics in *S. bryani*, it seems likely that the F element would be translocated to the X chromosome in this species. However, confirmation of this and other arm translocations will only come from future in situ hybridizations.

## 5. Conclusions

The *Scaptodrosophila* species we have examined indicate that, although this genus separated within the drosophilid lineage relatively early, the types of karyotypic change are similar to those observed in *Drosophila* species and other features are also shared (Table 4). Heterochromatin amplification appears to have been an important mechanism for structurally changing the arm length of the sex and ‘dot’ chromosomes in both *Scaptodrosophila* and *Drosophila*. Such amplification apparently adds arms to other chromosomes as well [12]. In *Drosophila* species, heterochromatin amplification seems to have been concentrated in specific groups such as members of the *ananassae* species subgroup [35] and several Hawaiian groups [48] where the mechanism and basis of this process have been discussed. In *Scaptodrosophila*, the amplification seems to be particularly important in the *coracina* species group. However, not enough species have been examined by modern banding techniques to be definitive about this. In the *Drosophila* karyotype list compiled by Clayton [49], a number of *Scaptodrosophila* karyotypes do not show a ‘dot’ (F) chromosome. One reason for this could be the presence in some species of a large ‘dot’ that was not recognized as heterochromatic by the techniques used. Heterochromatin amplification could be a widespread mechanism of karyotype evolution in the *Scaptodrosophila*. It is present in *S.* sp. *aff. concolor* strain CBR1, potentially a member of the *barkeri* group, and in *S. hibisci*, one of the flower-breeding species, where it could be related to neo-Y formation and male recombination. The locations of the NOR are the same in both *Scaptodrosophila* and *Drosophila* [35], as are the various mechanisms responsible for changing arm configurations. The many weak points and apparent inverted repeats observed in polytene chromosomes of *Scaptodrosophila* species may indicate a higher number of transposable elements in this group, but proof of this would require more investigation at the DNA level.

Of the estimated 230 described species of *Scaptodrosophila* [50], relatively few have been the subject of genetic studies. Preliminary whole genome DNA sequencing of some of these species (Rahul Rane, personal communication) has placed *S.* sp. *aff. concolor* strain CBN17 as a relatively recent divergence from the line leading to *coracina* species in the *Scaptodrosophila* tree. *S.* sp. *aff. concolor* CBR1 was not examined in these DNA comparisons, but the difference between the chromosomes of the two species suggests that, although the flies may look superficially similar to *S. concolor,* they may not be closely related and could actually belong to different species groups. The line leading to *S*. *xanthorrhoeae* diverges earlier in the DNA sequence comparisons, while *S. bryani* and *S. novoguineensis* form a separate and even earlier branching. *Drosophila lebanonensis* has been removed from the *Scaptodrosophila* lineage. The current analysis should provide a cytological framework for these ongoing genomic efforts.

## Figures and Tables

**Figure 1 insects-13-00364-f001:**
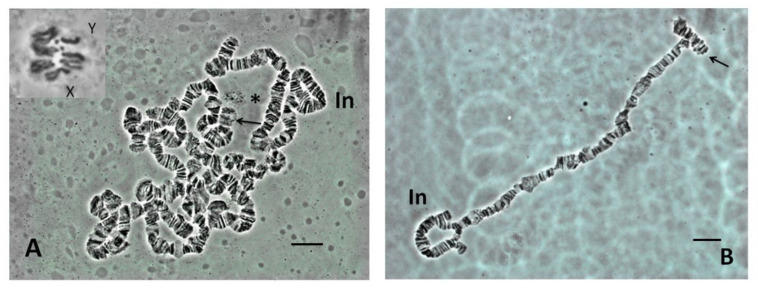
*Scaptodrosophila* sp. *aff. concolor* strain CBN17 salivary gland squash preparation. (**A**) Polytene nucleus. In = terminal inversion; arrow = ‘dot’ chromosome; * = nucleolar organizing region (NOR). Insert in upper left = mitotic set, X and Y = sex chromosomes. (**B**) Extended polytene chromosome. In = terminal inversion; arrow = inverted repeat. Bar = 10 µm.

**Figure 2 insects-13-00364-f002:**
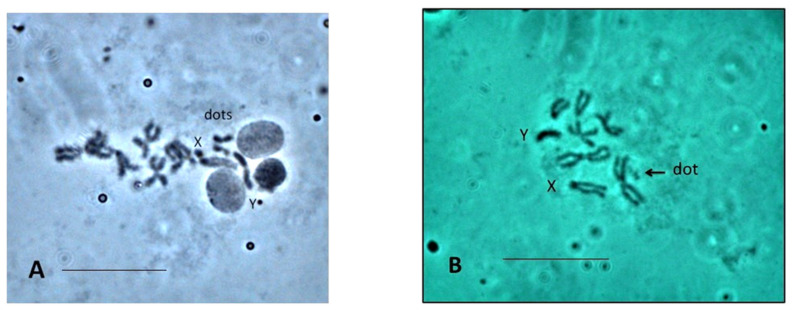
Cerebral ganglia squashes (**A**) *Scaptodrosophila* sp. *aff. concolor* strain CBR1, orcein squash, ‘dots’, X and Y indicated. (**B**) *Scaptodrosophila xanthorrhoeae*, orcein squash, ‘dot’, X and Y indicated. Bar = 5 µm.

**Figure 3 insects-13-00364-f003:**
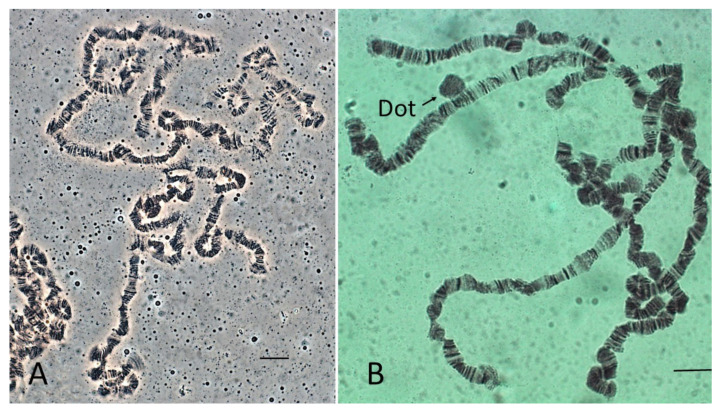
(**A**) *Scaptodrosophila bryani* polytene set. (**B**) *Scaptodrosophila cancellata* polytene set. Presumed ‘dot’ indicated. Bar = 10 µm.

**Figure 4 insects-13-00364-f004:**
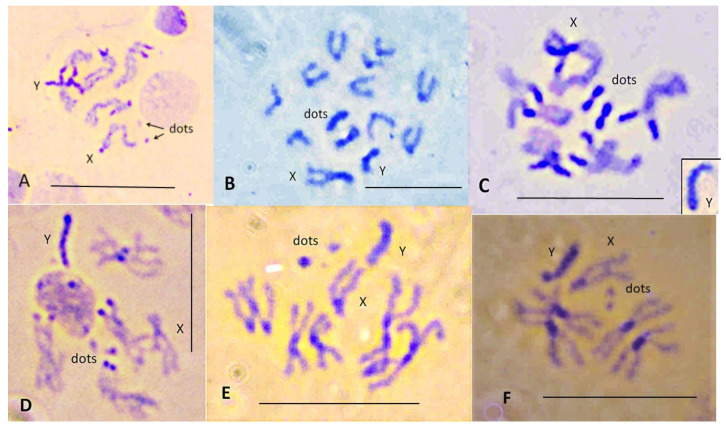
C-banding of cerebral ganglia chromosomes. (**A**) *Scaptodrosophila claytoni* male. X shows 2 internal C-bands. Y and ‘dots’ are also indicated. (**B**) *Scaptodrosophila* sp. *aff. cancellata* male, where X, Y and ‘dots’ are indicated. (**C**) *Scaptodrosophila cancellata* female, where X and ‘dots’ are indicated; Y is shown as an insert. (**D**) *Scaptodrosophila* sp. *aff. concolor* strain CBN17 male, note the pale C-band on X. (**E**) *Scaptodrosophila xanthorrhoea* male. (**F**) *S.* sp. *aff. novoguineensis* male. Bar = 5 µm.

**Figure 5 insects-13-00364-f005:**
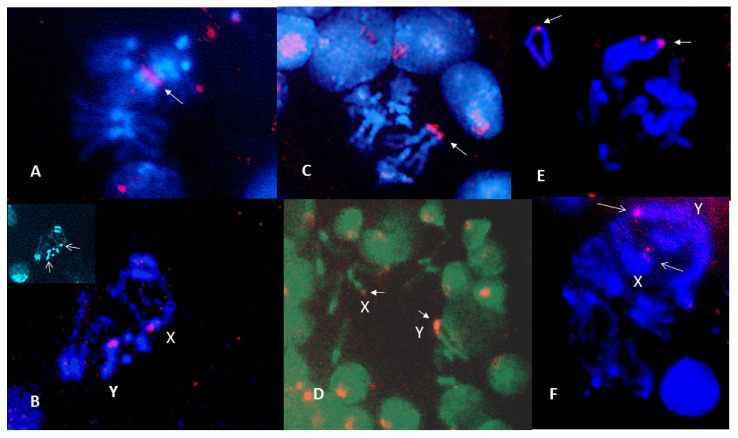
NOR sites shown by fluorescence in situ hybridization (FISH). Chromosomes stained with DAPI. (**A**) *Scaptodrosophila xanthorrhoea* female with NOR site on X chromosome (arrow). The rRNA technique was used. (**B**) *Scaptodrosophila xanthorrhoea* male with NOR sites on ends of X and Y chromosomes. Insert in upper left shows same photo with DAPI staining. Each chromosome or chromosome pair can be distinguished by the shape of its heterochromatin. Y (arrow) is completely heterochromatic. X has a single round heterochromatic end (arrow). An rDNA probe was used. (**C**) *Scaptodrosophila cancellata* female with NOR site at end of X chromosome heterochromatic arm (arrow). The rRNA technique was used. (**D**) *Scaptodrosophila cancellata* male set. Heterochromatic Y has NOR site at one end (arrow). X has pale NOR site at end of heterochromatic arm (arrow). An rDNA probe was used. (**E**) *Scaptodrosophila* sp. *aff. cancellata* female with site at end of heterochromatic arm of X (arrows). An rDNA probe was used. (**F**) *Scaptodrosophila* sp. *cancellata* male with strong site on heterochromatic Y chromosome (arrow) and weaker site at heterochromatic end of X (arrow). An rDNA probe was used.

**Figure 6 insects-13-00364-f006:**
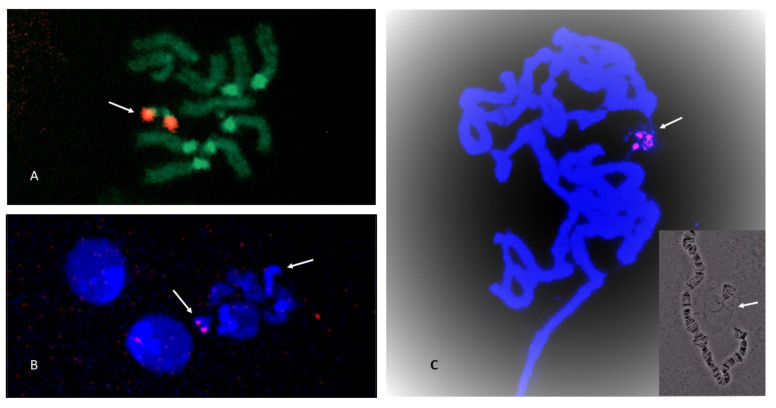
NOR sites on *Scaptodrosophila* sp. *aff. concolor* strain CBN17 chromosomes as shown by FISH. (**A**) Female with hybridization to ‘dot’ chromosomes (arrow). The rRNA technique was used. (**B**) Male with hybridization to ‘dot’ chromosomes. Second arrow indicates heterochromatic Y chromosome. An rDNA probe was used. (**C**) Polytene chromosomes with hybridization to nucleolar RNA. The rRNA technique was used. Insert shows close association between polytene ‘dot’ chromosome and nucleolus with extending strands of DNA.

**Figure 7 insects-13-00364-f007:**
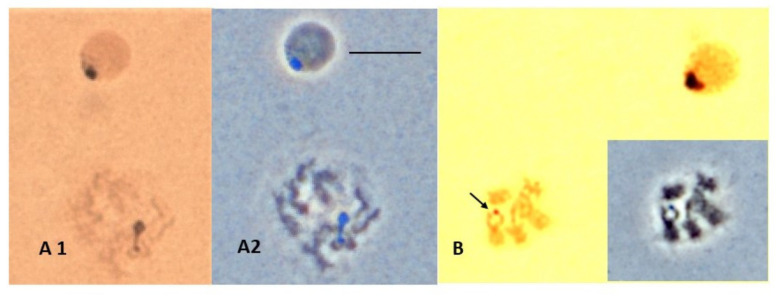
Silver staining technique showing *Scaptodrosophila* sp. *aff. cancellata* chromosomes. (**A**) Interphase and early mitotic cells with nucleolar staining. (**A1**) Bright field. (**A2**) Phase with distinctive coloration of nucleolus. Chromosomal origin of nucleolar fragment unknown. (**B**) Mitotic cell with silver-positive spot over apparent X chromosome. Bar = 5 µm. Insert shows phase contrast of the same cell.

**Figure 8 insects-13-00364-f008:**
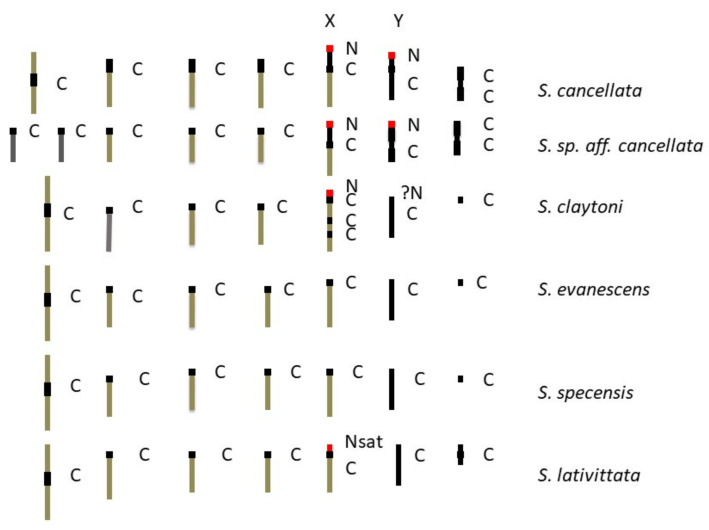
Ideograms of chromosomes examined. C = C-banded heterochromatic region; N = NOR location by FISH; ?N = possible NOR (FISH not conducted for male); Nsat = NOR identified by satellite location; N* = FISH conducted for *S.* sp *aff*. novoguineensis.

**Figure 9 insects-13-00364-f009:**
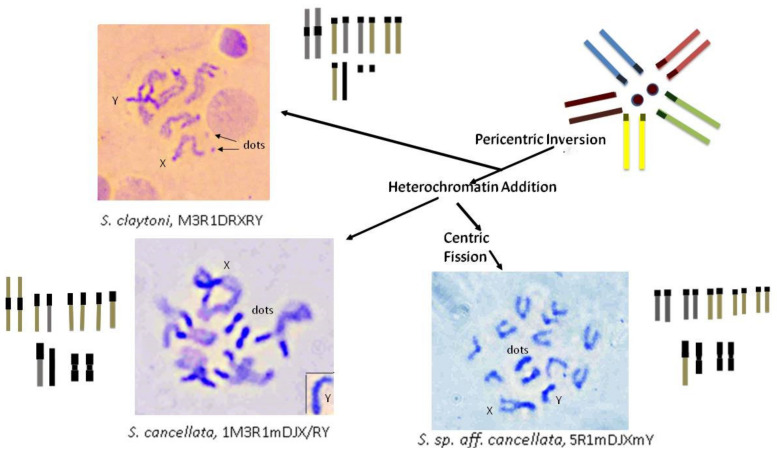
A scheme showing possible chromosome changes from an ancestral five acrocentric plus dot karyotype that could have given rise to members of the *coracina* group of species.

**Figure 10 insects-13-00364-f010:**
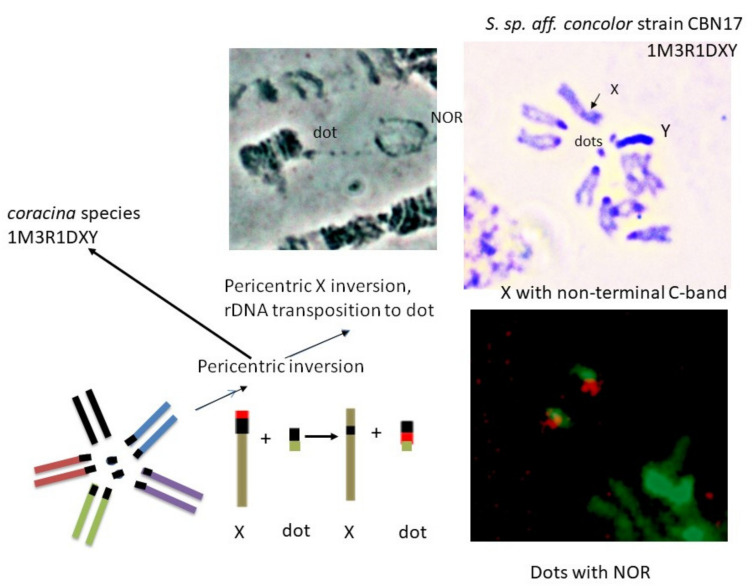
Scheme of changes in an ancestral karyotype that could have resulted in the formation of the *S.* sp. *aff. concolor* strain CBN17 karyotype.

**Figure 11 insects-13-00364-f011:**
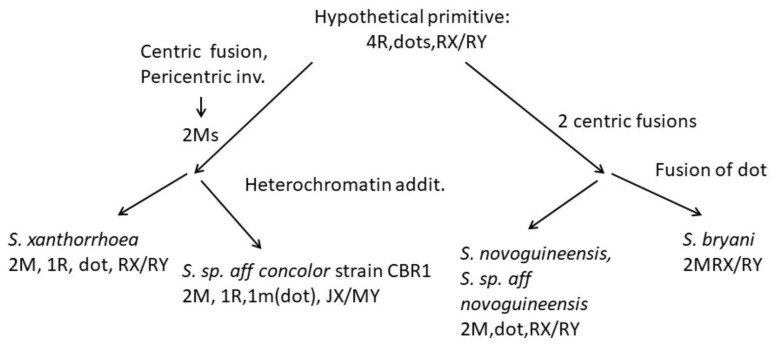
Possible karyotypic changes giving rise to other *Scaptodrosophila* species studied.

**Table 1 insects-13-00364-t001:** List of species and collection site of species examined in this study.

Species	Species Group	Collection Site
*Scaptodrosophila cancellata*	*coracina*	Lake Placid, Qld (September 2011)
*Scaptodrosophila* sp. *aff. cancellata*	*coracina*	Lake Placid, Qld (April 2012)
*Scaptodrosophila claytoni*	*coracina*	Nowra, NSW (March 2014)
*Scaptodrosophila evanescens*	*coracina*	Nowra, NSW (March 2014)
*Scaptodrosophila specensis*	*coracina*	Lake Placid, Qld (April 2012)
*Scaptodrosophila lativittata **	*coracina*	Melbourne, Vic (March 2021)
*Scaptodrosophila nitidithorax*	*coracina*	South Perth, WA (February 2021)
*S.* sp. *aff. concolor* strain CBN17	*barkeri?*	Townsville, Qld (September 2011)
*S.* sp. *aff. concolor* strain CBR1	*barkeri?*	Lake Placid, Qld (September 2011)
*Scaptodrosophila xanthorrhoeae*		Lake Placid, Qld (September 2011)
*Scaptodrosophila novoguineensis*		Mossman, Qld (May 2011)
*Scaptodrosophila* sp. *aff. novoguineensis*		Lake Placid, Qld (April 2013)
*Scaptodrosophila bryani*	*bryani*	Lake Placid, Qld (May 2011)

* Collected and identified by Dr. Belinda van Heerwaarden.

**Table 2 insects-13-00364-t002:** Methods used.

Species	Chromosomes Examined	Techniques
*Scaptodrosophila cancellata*	ganglion	orcein staining, C-banding,
		rDNA FISH
	polytene	squash preparation
*S.* sp *aff cancellata*	ganglion	orcein staining, C-banding,
		rDNA FISH
	polytene	squash preparation
*S. claytoni*	ganglion	orcein staining, C-banding, rDNA FISH (female)
	polytene	squash preparation
*S. evanescens*	ganglion	orcein staining, C-banding
	polytene	squash preparation
*S. specensis*	ganglion	C-banding
	polytene	not done
*S. lativittata*	ganglion	orcein staining, C-banding, NOR from satellites
	polytene	squash preparation
*S. nitidithorax **	polytene	squash preparation
*S.* sp. *aff concolor* strain CBN17	ganglion	orcein staining, C-banding,
		rDNA FISH
	polytene	squash preparation
*S.* sp. *aff concolor* strain CBR1	ganglion	orcein staining
	polytene	squash preparation
*S. xanthorrhoeae*	ganglion	orcein staining, C-banding
		rDNA FISH
	polytene	squash preparation
*S. novoguineensis*	ganglion	orcein staining, C-banding
	polytene	squash preparation
*S.* sp. *aff novoguineensis*	ganglion	orcein staining, C-banding
		rDNA FISH
	polytene	squash preparation
*S. bryani*	ganglion	orcein staining
	polytene	squash preparation

* Ganglion chromosomes examined by Bock [11].

**Table 3 insects-13-00364-t003:** Karyotypes of the species (M = large metacentric; m = small metacentric; J = submetacentric; R = rod or acrocentric; D = dot) * from Bock [11], and # from Wilson et al. [12].

Species	Female Brain	Male Brain	Total Chromosomes, Female
*S. cancellata*	1M3R1mDJX	1M3R1mDJXRY	6 pairs = 9 arms
*S.* sp. *aff. cancellata*	5R1mDJX	5R1mDJXmY	7 pairs = 9 arms
*S. claytoni*	1M3R1DRX	1M3R1DRXRY	6 pairs = 7arms
*S. evanescens*	1M3R1DRX	1M3R1DRXRY	6 pairs = 7arms
*S. specensis*	1M3R1DRX	1M3R1DRXRY	6 pairs = 7arms
*S. lativittata*	1M3R1mDRX	1M3R1mDRXRY	6 pairs = 8 arms
*S. nitidithorax* *	1M3R1JDRX	1M3R1JDRXRY	6 pairs = 8 arms
*S. enigma* *	1M3R1MDJX	1M3R1MDJXJY	6 pairs = 9 arms
*S. howensis **	1M3R1MDJX	1M3R1MDJXJY	6 pairs = 9 arms
*S. novamaculosa **	1M3R1JDJX	1M3R1JDJXRY	6 pairs = 9 arms
*S.* sp. *aff. concolor* strain CBN17	1M3R1DJX	1M3R1DJXRY	6 pairs = 8 arms
*S.* sp. *aff. concolor* strain CBR1	2M1R1mDJX	2M1R1mDJXMY	5 pairs = 9 arms
*S. xanthorrhoeae*	2M1R1DRX	2M1R1DRXRY	5 pairs = 7 arms
*S. novoguineensis*	2M1DRX	2M1DRXRY	4 pairs = 6 arms
*S.* sp. *aff. novoguineensis*	2M1DRX	2M1DRXRY	4 pairs = 6 arms
*S. bryani*	2MRX	2MRXRY	3 pairs = 5 arms
*S. hibisci* ^#^	1M3m1J1D ^$^	1M3m1J1D ^$^	6 pairs = 10 arms

^$^ Sex chromosomes not distinguished by the authors.

**Table 4 insects-13-00364-t004:** Comparison of *Scaptodrosophila* with *Drosophila*.

Characteristic	Observations in *Drosophila*	Observations in *Scaptodrosophila*
Link to ancestral karyotype	Yes	Not observed
Heterochromatin location	Centromeres, ‘dots’, Y, some arms	As for *Drosophila*
NORs	X, Y usually, dot, occasionally others	X, Y, dot
Dots	Size varies, heterochromatic but C-banding variable, usually present in polytene set	Size varies, heterochromatic but C-banding variable, usually absent in polytene set
Mitotic chromosomes	A–F arms, syntenic	A–F arms, assumed
Polytene chromosomes	Generally spread well	Spread poorly, weak points, repeats

## Data Availability

All data are available within the manuscript.

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
