# Peer review of "Chromosome Comparisons of Australian Scaptodrosophila Species"

_insects, 2022, doi:10.3390/insects13040364_

Round 1

Reviewer 1 Report

Hello Dear 

The manuscript is well written in all its parts and its processing is done clearly and appropriate references are provided throughout the version. Hence, I recommend publishing the manuscript in insects.

Author Response

Thank you to this reviewer.

Reviewer 2 Report

In their work “Chromosome Comparisons of Australian Scaptodrosophila Species“ the authors characterize the karyotypes of 12 species belonging to the Scaptodrosophila genus. To that purpose they perform C and NOR bandings, propose pathways of karyotype evolution and compare it to the karyotypes observed within Drosophila genus.

The manuscript is well written and clear in all its sections, with appropriate references provided throughout the manuscript. Thereby, I recommend the publication of the manuscript in Insects. My comments and suggestions for the authors are listed below:

(Pages and lines of the manuscript are not numbered, making it harder to refer to a specific part of the manuscript.)

I suggest the information brought in paragraph starting with “Polytene chromosomes of S. nitidithorax were examined….” (page 5 of the pdf file) to be presented in form of a table for better overview, listing the species, type of preparation and type of banding performed.

I suggest an ideogram of the chromosomes of the inspected species showing similarities and differences in karyotypes and in banding patterns (C, NOR) to be introduced, prior to the schemes brought in Figures 8, 9 and 10. In this way the discussed differences could be followed more easily.

Some reference should be provided for this sentence: “For S. lativittata, which was not considered in the in situ hybridizations, the presence of distinctive satellites at the heterochromatic end of the X chromosomes indicated that this was one of the NOR sites.”

Is the term “normal” necessary in the sentence “However, their repeat level is considerably lower than pericentric heterochromatin since normal gene sequences are dispersed within them (Riddle and Elgin 2018).”?

“In Drosophila, karyotype examination (Muller 1940, Sturtevant and Novitski 1941) and more recent sequencing…” To which species of the Drosophila genus this refers to?

In the conclusion the authors mention "DNA comparisons" and "DNA sequence comparisons" when discussing species separation. It would be good to mention which type of sequence analysis yielded those results/conclusions. Phylogenetic analysis based on COI, or on ribosomal genes...? Or some other type of analysis?

Throughout the manuscript “in situ” should be in italic.

Full name for the NOR abbreviation has not been brought at the beginning of the manuscript. Later throughout the text both „NOR” and „nucleolar organizer region” are being used. On the first mention please bring the full name and the abbreviation and later in the text use only the abbreviation.

Author Response

Please see the attachment in the box. Thank you for your comments and suggestions.

Reviewer 3 Report

In this paper, the authors studied the mitotic karyotypes of 12 Scaptodrosophila species. In some species, also polytene chromosomes were examined. Mitotic chromosomes C-banding and NOR location by FISH data were used to analyzed the possible chromosome evolution in this species group.

The results are interesting and provide new data for understanding the evolution of this genus. However, some shortcomings must be solved before acceptance of the manuscript. In addition, I have also minor suggestions in order to improve the manuscript.

The section “Material and Methods” must be improved, specially the “in situ hybridization” section. The authors detail how the probes are labeled and how hybridization is carry out. However, they do not indicate how hybridization detection is performed.

Two different FISH probes are used. The first one was obtained from a plasmid containing an rDNA unit from Drosophila melanogaster. An alternative RNA probe was prepared using total RNA. The use of cloned or PCR-derived DNA sequences to generate probes is common in cytogenetics laboratories. The use of RNA is not very common for the localization of NORs. Why do the authors use this technique? Do they obtain the same results with both techniques? Although the RNA is enriched in ribosomal RNA there are also other RNAs, are nonspecific hybridizations possible?

The authors use a karyotype nomenclature similar to the one used for Drosophila karyology by Clayton & Wheeler (1975). This nomenclature is difficult to follow for readers unfamiliar with Drosophila cytogenetics. It might be useful to explain it for a broad audience of potential readers of "Insects". Anyway, I have reviewed the paper of Clayton and Wheeler as well as other studies on Drosophilidae cytogenetics and the mitotic formulae do not include information on the sex chromosomes, as the authors do. This can be confusing. For example, the karyotype for S. hibisci is 1M3m1J1D, without information about sex chromosomes. I understand that the sex chromosomes are not identified in this species but this point must be clarified.

It would have been nice if in Figures 8, 9 and possibly 10, the authors use karyotype schemes in the same way as is done for the ancestral karyotype, instead of repeating photos already included in other figures. It would be easier to understand the karyotypic evolution in these species. I suggest the authors do something similar to what is presented in Figure 3 in Craddock et al. (2016). In any case, it would be good to include in Figure 8 the karyotype formulae, as it is done in Figure 9.  In Figures 9 and 10 it would be convenient to include in the karyotypes formulae the information of the sex chromosomes. In Figure 10 please change “V” by “M” to avoid confusions.

It would have been better if the authors had used the “template file” of the journal or at least used numbered lines. This would have simplified the review.

Minor:

The images in Figure 2 should be improved by increasing the contrast.

A scale bar should be included in all images to estimate the chromosome size.

Page 4, first paragraph. In this paragraph, they are talking about results. Move to the “results “section

Page 7: 19S?

Page 7:  Change “Madelena” by “Madalena”

Page 8. The first three lines are material and methods. Delete

Table 2:  S. cancellata, 1M3R1mD1JXRY or 1M3R1mDJXRY?

Table 2: the numbers of chromosome arms is for the male or the female karyotype? This could change according to the sex chromosome morphology.

Although "rod" and "acrocentric" are similar, the use of both terms throughout the manuscript can sometimes be confusing. The same for “metacentric” and “V-shaped”. In the figure 10 is used the abbreviation “V” (V-shaped chromosomes) that is equivalent to “M” (metacentric) that is used in other parts of the manuscript as the Table 2. Please use only one of them

Page 12, last paragraph. It is well known that Ag-staining is not useful to determinate the rDNA loci positions or estimate their activity in all species. In many species silver stains other regions different to the NOR, usually heterochromatic regions. This has been observed in some mammals, fishes and amphibians as well as in many insect species as ants, wasps, beetles or grasshoppers.

Author Response

Please see the attachment. Thank you for your comments and suggestions.

Round 2

Reviewer 3 Report

The manuscript has been greatly improved. I am really happy with the new Figures 8, 9 and 10. The modifications make it much easier to follow the discussion of the work.

I have only minor suggestions indicated directly in the manuscript.

Author Response

I have made all the suggested corrections. (Please see the attachment.) Thank you for your careful examination. (I hope I am returning the corrected manuscript!) 
